

# Phylogeography and population genetic structure of the cardinal tetra (*Paracheirodon axelrodi*) in the Orinoco basin and Negro River (Amazon basin): evaluating connectivity and historical patterns of diversification

Diana Sanchez-Bernal[1,*], José Gregorio Martinez[2,*], Izeni Pires Farias[3], Tomas Hrbek[3] and Susana Caballero[1]

[1] Biological Sciences Department, Universidad de Los Andes, Bogota, Colombia
[2] Grupo de Investigación Biociencias, Facultad de Ciencias de la Salud, Institución Universitaria Colegio Mayor de Antioquia, Medellín, Antioquia, Colombia
[3] Departamento de Genética, Universidade do Amazonas, Manaos, Brazil
[*] These authors contributed equally to this work.

Corresponding author
Susana Caballero,
sj.caballero26@uniandes.edu.co

## ABSTRACT

The Neotropics contain one of the most diverse assemblages of freshwater fishes worldwide. Part of this diversity is shared between the Orinoco and Amazon basins. These basins have been separated for a long time due to the Vaupes Arch, rising between 10–11 Ma. Today, there is only one permanent connection between the Orinoco and Negro (Amazon) basins, known as the Casiquiare Canal. However, alternative corridors allowing fish dispersion between both basins have been proposed. The cardinal tetra (*Paracheirodon axelrodi),* the most important fish in the ornamental world market, is distributed in both basins. Here we investigated *P. axelrodi* phylogeography, population structure, and potential routes of migration and connectivity between the two basins. A total of 468 bp of the mitochondrial gene (COI), 555 bp of the nuclear gene fragment (MYH6), and eight microsatellite loci were analyzed. As a result, we found two major genetic clusters as the most likely scenario ($K = 2$), but they were not discreetly distributed between basins. A gradient of genetic admixture was observed in Cucui and São Gabriel da Cachoeira, between the upper Negro River and the upper Orinoco. Samples from the middle-lower Negro River were highly structured. Cucui (Negro basin) was more similar to the Orinoco than to the rest of the Negro basin populations. However, substructure was also observed by the discriminant analysis, fixation indices and other hierarchichal structure analyses ($K = 3 - 6$), showing three major geographic clusters: Orinoco, Cucui, and the remaining Negro basin. Unidirectional migration patterns were detected between basins: via Cucui toward Orinoco and via the remaining of the Negro basin toward Orinoco. Results from the Relaxed Random Walk analysis support a very recent origin of this species in the headwater Orinoco basin (Western Guiana Shield, at late Pleistocene) with a later rapid colonization of the remaining Orinoco basin and almost simultaneously the Negro River via Cucui, between 0.115 until about 0.001 Ma. Historical biogeography and

population genetic patterns observed here for Cardinal tetra, seem to be better explained by river capture, physical, or ecological barriers than due to the geographic distance.

## INTRODUCTION

The Neotropics contain one of the most diverse assemblages of freshwater fishes in the world (*Reis, Kullander & Ferraris, 2003*; *Turner et al., 2004*; *Albert, Tagliacollo & Dagosta, 2020*). Within the Neotropics, the Orinoco and Amazon basins are characterized by their high ichthyofauna diversity, exceeding 9,000 total species, of which ∼5,160 are freshwater species (*Reis et al., 2016*; *Albert, Tagliacollo & Dagosta, 2020*).

The origins of this ichthyofauna date back to the middle Miocene where most of the area of what is now the western Amazon comprised the "Paleo Amazonas-Orinoco" drainage. It is believed that this system drained northward into the Magdalena River, and, after the rise of the Cordillera Oriental and Sierra de Perijá, into Lake Maracaibo (*Hoorn, 1994*; *Hoorn et al., 1995*; *Díaz De Gamero, 1996*). This ancient river drained areas that are now part of the Magdalena, Orinoco, and Amazonas River basins, implying an extensive, biogeographically interconnected region (*Albert, Lovejoy & Crampton, 2006*; *Winemiller & Willis, 2011*).

The rise of the Vaupes Arch in the late Miocene (between 10–11 Ma) severed this connectivity, resulting in the formation of two basins: the Orinoco, running north and northeast, and the Amazon, flowing towards the east, until it reached its current delta in the Atlantic Ocean, once the Purús Arch ruptured (*Hoorn et al., 1995*; *Díaz De Gamero, 1996*; *Lundberg et al., 1998*; *Hubert & Renno, 2006*; *Winemiller & Willis, 2011*; *Mora et al., 2011*). At present, there is only one permanent connection between the two basins: the Casiquiare Canal. The Casiquiare Canal originated by a partial headwater capture of the upper Orinoco River, with the southern branch emptying into the upper Negro River, the largest tributary of the Amazon River, and the northern branch continuing as the Orinoco River (*Winemiller et al., 2008*).

Along its course, the Casiquiare Canal has a gradient of water types, ranging from clear water (waters low in organic matter and sediment) near its source in the upper Orinoco, to black waters (waters rich in organic matter but low in sediments) near its confluence with the Negro River (*Sioli, 1984*; *Crampton, 2011*). This environmental gradient appears to act as an ecological filter for several fish species, while the canal itself allows for connectivity between the ichthyofaunas of the Orinoco and the Amazon basins (*Winemiller et al., 2008*).

In addition to the Casiquiare Canal, alternative corridors allowing fish movement between the Orinoco and Amazon basins have been proposed (*Winemiller et al., 2008*; *Escobar et al., 2015*). One of these corridors between the two basins is *via* the Essequibo River and the Rupununi savanna during the high-water season. This alternative corridor has been used to explain the dispersion of the ancestors of marine species that colonized the freshwater environments of the two basins (*Lovejoy & Araujo, 2000*), and to explain

the biogeographical processes behind the diversification of species between the two basins (*Escobar et al., 2015*).

Although the Orinoco and Amazon basins have been separated for a long time, these two basins share a large number of species (*Reis, Kullander & Ferraris, 2003*; *Escobar et al., 2015*). One such species is *Paracheirodon axelrodi,* commonly known as the cardinal tetra. This fish is distributed in the upper Orinoco basin and the upper and middle Negro basin in the Amazon Basin. This species uses a flooded forest environment for feeding and reproduction. During the flooding cycle they move from small streams to the flooded forest, returning to small streams during low water season. These dynamics of flood cycles provide conditions for migration and genetic exchange among populations (*Cooke, Chao & Beheregaray, 2009*).

The cardinal tetra is the most important ornamental fish in the world market (*Chao, 2001*; *Terencio, Schneider & Porto, 2012*) comprising more than 80% of the total ornamental fish exported from the Negro River each year (*Chao, 2001*; *Beheregaray et al., 2004*; *De Oliveira et al., 2008*). Moreover, it is one of the most commercialized species in the Orinoco basin, ranking in the first place between the years 1998 and 2004 (*Tovar et al., 2009*). Despite its extensive geographic distribution spanning two major South American river basins, and extensive commercial exploitation, very little is known about its population dynamics, population structuring or evolutionary history of this species using molecular markers.

The first molecular study reported for *P. axelrodi* was that of *Harris & Petry (2001)*. The authors analyzed the mitochondrial control region for specimens from two geographically proximate localities from the vicinity of Barcelos (Brazil) and specimens of unknown geographic origin and found 3.9% and 4.3% sequence divergence between these specimens. In a much more comprehensive study focusing on the Negro River, *Cooke & Beheregaray (2007)* found extremely high levels of genetic diversity in the S72 intron in specimens sampled from the tributaries of the Negro River. Subsequently, *Cooke, Chao & Beheregaray (2009)* evaluated population structure, history of colonization, and genealogical relationships, concluding that headwater populations showed higher genetic diversity than downstream populations. *Bittencourt et al. (2017)* analyzed temporal genetic variation within a single population using five microsatellite loci and concluded that changes in allelic frequencies over time are associated with droughts instigated by El Niño, although these events do not significantly reduce genetic diversity. Despite these studies, none has so far revealed the evolutionary behavior of the Orinoco population and its relationship to the Amazon populations, combining information from mitochondrial and nuclear markers.

In this study we provide the first phylogeographic hypothesis and analysis of population structure for *P. axelrodi* in its entire geographic distribution in the Orinoco and Amazon basins using different molecular markers, including mitochondrial (COI), nuclear sequences (MYH6), and eight microsatellite loci. Additionally, we test the hypothesis of basins connectivity and provide support regarding potential routes of gene flow and colonization between the two basins.

## MATERIALS & METHODS

### Sample collection and DNA extraction

A total of 163 muscle tissue samples (one per individual) were collected along the Orinoco and Amazon basins on the distribution area for *P. axelrodi.* The samples were collected by ornamental fish fishermen from four localities on the Orinoco basin (Colombia) including San José del Guaviare ($n = 20$), Puerto Carreño ($n = 20$), Puerto Gaitan ($n = 21$), and Puerto Inirida ($n = 21$) (Fig. 1), and four localities in Negro River in the Amazon basin (Brazil) including Cucui ($n = 21$), Santa Isabel ($n = 20$), Barcelos ($n = 20$), and *São Gabriel da Cachoeira* ($n = 20$) (Fig. 1). Samples from São Gabriel da Cachoeira (Negro River–Amazon basin) were obtained from the Laboratório de Genética Animal (LGA) at Instituto Nacional de Pesquisas da Amazônia (INPA). However, these samples could not be used for mitochondrial and nuclear DNA analysis due to the low integrity of DNA, but they were suitable for microsatellite analysis. Samples from all other Amazon localities were also obtained from local artisanal fishermen working in these areas selling ornamental fish. Tissue samples and DNAs were deposited on the Laboratorio de Ecología Molecular de Vertebrados Acuáticos–LEMVA at Universidad de los Andes, Colombia. Details on sample information and depositaries can be consulted in the Table S1.

Once the fish were received, they were euthanized by rapid chilling (hypothermic shock) on ice, following the American Veterinary Medical Association (AVMA) guidelines for the euthanasia of animals: 2020 Edition (https://www.avma.org/resources-tools/avma-policies/avma-guidelines-euthanasia-animals). This procedure is recommended for small-bodied (3.8-cm-long or smaller) tropical and subtropical stenothermic fish like cardinal tetra. Then, all individuals and tissue samples were preserved in ethanol at 95% and stored at 4 °C. Genomic DNA was extracted using MoBio UltraClean® Tissue and Cells DNA Isolation Kit or following a phenol/chloroform protocol (*Sambrook, Fritsch & Maniatis, 1989*).

### PCR amplification and sequencing
#### Mitochondrial DNA

In this study, 96 mitochondrial fragments of the cytochrome oxidase I (COI) gene corresponding to 468 base pairs (bp) length each one (7–21 individuals per locality), were amplified using the primers designed by *Ivanova et al. (2007)*. PCR reactions were carried out in a final volume of 15 μL and contained 8.3 μL of ddH$_2$O, 1.2 μL of MgCl$_2$ (25 mM), 1.5 μL of 10x buffer (($NH_4$)$_2$SO$_4$), 1.2 μL de dNTPs (10 mM), 1.5 μL of both primers (2 μM), 0.3 μL of Taq DNA Polymerase (1 U/ μL) and 1 μL of DNA (∼50 ng/μL). PCR conditions were as follows: 72 °C for 1 min, 35 cycles of 94 °C for 10 s, 50 °C for 35 s and 72 °C for 90 s, with a final extension period of 72 °C for 5 min.

#### Nuclear DNA

Nuclear genome sequences corresponding to 555 bp of exonic region myosin heavy polypeptide 6 cardiac muscle alpha (MYH6), were amplified only for samples of *P. axelrodi* representing unique haplotypes for COI gene within each locality, previously identified as described in the ''Phylogeographic analyses'' section. We used
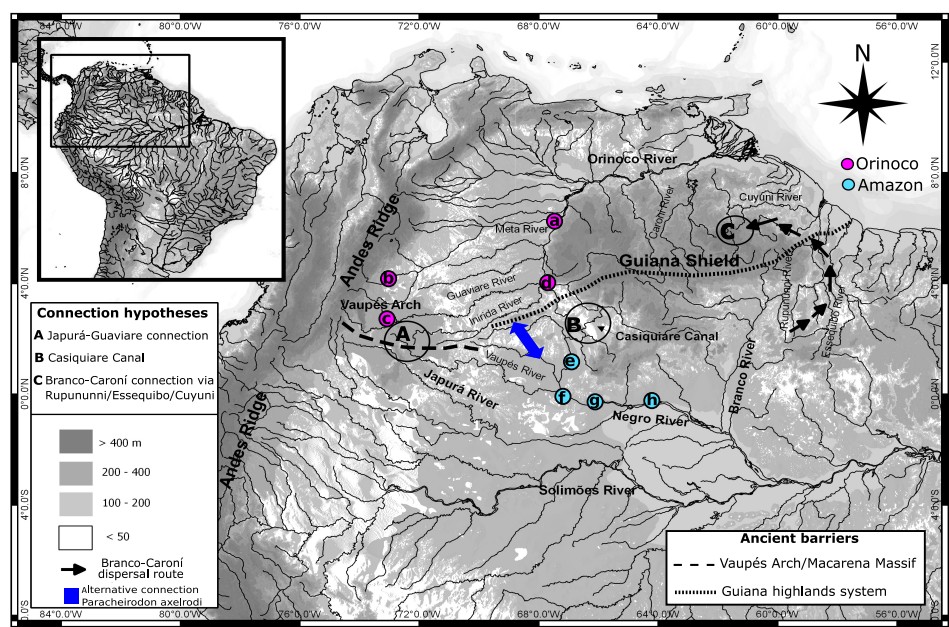

**Figure 1** **Map showing hydro-geological scenario and sampling areas.** Map showing the hydro-geological scenario of study and sampling localities of *Paracheirodon axelrodi*. Negro basin (Amazon) locations are in purple. Orinoco basin locations are in green. Locations are (A) Puerto Carreño (*n* = 20), (B) Puerto Gaitán (*n* = 21), (C) San José del Guaviare (*n* = 20), (D) Puerto Inirida (*n* = 21), (E) Cucui (*n* = 21), (F) São Gabriel da Cachoeira (*n* = 20),(G) Santa Isabel (*n* = 20), (H) Barcelos (*n* = 20). Points A, B and C represent the three more famous hypotheticals connectivity routes between Orinoco and Amazon basins for the ichthyofauna (*Lovejoy & Araujo, 2000*; *Winemiller & Willis, 2011*; *Escobar et al., 2015*). Alternative connection routes suggested in this study for the Cardinal tetra are represented in blue.

the primers designed by *Li et al. (2007)*. For nuclear gene (MYH6) amplification, nested PCR was performed. For this, two independent working solution of 2 μM were set up by combining equimolarly the forward and reverse primers for each gene: MYH6_F459 (5c CATMTTYTCCATCTCAGATAATGC 3′) with MYH6_R1325 (5′ATTCTCACCACCATCCAGTTGAA 3′). Then, the 1st PCR reaction was carried out in a final volume of 15 μL and contained 5.1 μL of ddH$_2$O, 1.5 μL of BSA (10 mg/mL), 1.2 μL of MgCl$_2$ (25 mM), 1.5 μL of 10x buffer ((NH$_4$)$_2$SO$_4$), 1.2 μL of dNTPs (10 mM), 1.5 μL of each primer working solution, 0.5 μL of Taq DNA Polymerase (1 U/ μL) and 1 μL of DNA (∼50 ng/ μL). The 2nd-PCR reaction was carried out in a final volume of 15 μL and contained 8.3 μL of ddH$_2$O, 1.2 μL of MgCl$_2$ (25 mM), 1.5 μL of 10x buffer ((NH$_4$)$_2$SO$_4$), 1.2 μL de dNTPs (10 mM), 1.5 μL of both primers (2 μM), 0.3 μL of Taq DNA Polymerase (1 U/ μL) and 1 μL of 1st PCR product. PCR reagents were obtained from Fermentas (Vilnius, Lithuania).

PCR conditions for nuclear genes were as follows: 1st PCR - 68 °C for 1 min, 30 cycles of 93 °C for 10 s, 48 °C for 35 s and 68 °C for 90 s, with a final extension of 68 °C for 7 min. 2nd–PCR: 68 °C for 1 min, 35 cycles of 93 °C for 10 s, 55 °C for 35 s and 68 °C for 90 s., with a final extension period of 68 °C for 7 min. PCR products were purified using EXO-SAP, (Exonuclease—Shrimp Alcaline Phosphatase) following the manufacturer's suggested

protocol (*Werle et al., 1994*). All PCR products (mtDNA and nuDNA) were sequenced on an ABI 3500 automated sequencer (Applied Biosystems) using the BigDye Terminator Kit Cycle Sequencing Kit (Applied Biosystems) according to the manufacturer's instructions. The assembly, alignment, and quality check of the nucleotide sequences were performed using Geneious (*Kearse et al., 2012*).

Eight polymorphic microsatellite loci (*Pa7, Pa13, Pa19, Pa26, Pa27, Pa32, Pa33, Pa37*) developed for *P. axelrodi* by *Beheregaray et al. (2004)* were used. PCR reactions were carried out in a final volume of 10 µL and contained 3.7 µL of ddH$_2$O, 1.0 µL of MgCl$_2$ (25 mM), 1.0 µL of 10x buffer((NH$_4$)$_2$SO$_4$) ,1.0 µL of dNTPs (10 mM), 1.0 µL of reverse primer (2 µM), 0.5 µL of forward primer (2 µM), 0.5 µL of FAM –labelled M13 primer (2 µM), 0.3 µL of Taq DNA Polymerase (1 U/ µL) and 1 µL of DNA ($\sim$25 ng/µL). PCR reagents were obtained from Fermentas (Vilnius, Lithuania). Amplification conditions were as follows: 94 °C for 1 min, 25 cycles of 94 °C for 30 s, 60 °C for 30 s and 68 °C for 40 s, 30 cycles of 94 °C for 20 s, 52 °C for 30 s and 72 °C for 1 min, with a final extension of 72 °C for 30 min. Then, 1 µL of PCR product was combined with one mL of ROX size standard (*DeWoody et al., 2004*) and eight mL of Hi-Di formamide. All microsatellite loci were amplified separately. They were then combined and genotyped on an Applied Biosystems ABI 3100 Genetic Analyzer (Universidade Federal do Amazonas—UFAM). Allele sizes were inferred using the pUC19 ROX-labelled size standard (*DeWoody et al., 2004*). Subsequently, a matrix of genotypes for each individual was generated.

## Phylogeographic analyses

All sequences were edited manually and aligned using the software Geneious v4.7 (*Kearse et al., 2012*). After the alignment by the ClustalW algorithm (*Thompson, Higgins & Gibson, 1994*), unique COI haplotypes within each population was inferred using the online fasta sequence toolbox DNAcollapser from FaBox v1.61 (https://birc.au.dk/~palle/php/fabox/palle/php/fabox/; (*Villesen, 2007*). Then, a haplotype network was constructed for each gene to determine ancestry relationships. In the case of the MYH6, sequences were first phased into individual alleles using the PHASE algorithm (*Stephens, Smith & Donnelly, 2001*) incorporated into DnaSP v5.1 software (*Librado & Rozas, 2009*). Then, we used a maximum likelihood approach in RAxML v8.2.X for phylogenetic analyses (*Stamatakis, 2014*) in CIPRES (*Miller, Pfeiffer & Schwartz, 2010*) because of its ability to efficient and fast maximum likelihood tree search algorithm that returns trees with good likelihood scores. We used a GTR+GAMMA model of sequence evolution, the default and unique available for RAxML analysis, according the authors of the program in the version 8.2.X manual, for single full ML tree searches. In addition, we generate 1,000 replicates of RAxML's rapid bootstrap algorithm to account for uncertainty in the estimation of the topology (*Stamatakis, 2014*). A consensus tree was constructed from the bootstrap output file without ruling out any of the 1,000 replicates, for the allele network visualization into Haploviewer software (*Salzburger, Ewing & Von Haeseler, 2011*), a haplotype viewer program specifically developed for the purpose of reconstructing haplotypes networks using traditional phylogenetic algorithms, neighbour-joining, maximum parsimony and

maximum likelihood genealogies from closely related, and hence, highly similar haplotype sequence data.

As a final analysis, we used non-concatenated COI and MYH6 sequences in a unique phylogeographic analysis with a Bayesian statistical approach based on a relaxed random walk model in continuous space and time in BEAST v1.8.2 software (*Drummond et al., 2012*), on the CIPRES platform (*Miller, Pfeiffer & Schwartz, 2010*). This analysis was used for the reconstruction of the origin and dispersal routes of *P. axelrodi* between Amazon and Orinoco basins. This method infers evolutionary history in a continuum landscape combining DNA sequences with geographic coordinates, generating a genealogy, and estimating the ancestral localities of the internal nodes, taking into consideration uncertainties in topology (*Lemey et al., 2010*).

The models of substitution selected for the Bayesian analysis were HKY+GAMMA for COI and, GTR+GAMMA for MYH6. Both substitution models were previously estimated in JModelTest (*Posada, 2008*). As tree prior, a relaxed non-correlated model with a relaxed molecular clock was used, with the tree coalescence prior of the Bayesian Skyline. For the time calibration of the resulting tree, the geometric median derived from a substitution rate of $0.68 \times 10^{-8}$ mutations per site per year for COI (from the widely accepted mtDNA substitution rate for poikilotherm vertebrates in *Martin & Palumbi (1993)*, and $1.0 \times 10^{-9}$ for MYH6 (*Freeland, 2005*). Two independent runs were performed with a chain length of a 100 million, with a log every 10,000 steps. Run convergence was evaluated in Tracer v1.6.0 considering the stationary behavior of the chains and the effective sample size (ESS) > 300. Summary of the topologies to generate a clade credibility tree, was done on TreeAnnotator v1.8.2. Then, the two independent runs were combined. To generate space and time projections of the genealogies, keyhole markup language (kml) was used in the software SpreaD3 v0.9.6 (*Bielejec et al., 2016*), and the results were accessed and visualized using Google Maps.

## Gene flow and population genetic structure analyses

To establish the genetic structure and connectivity between the Orinoco and Negro River populations, a matrix for microsatellite allele was initially constructed in GeneMapper v4.1 (Applied Biosystems). Null allele frequencies were estimated in FreeNA Software (*Chapuis & Estoup, 2007*).

Arlequin v3.5 (*Excoffier & Lischer, 2010*) was used to run an Analysis of molecular variance AMOVA and Pairwise *Fst* among localities. Additionally, the number of alleles per locus ($N_A$), the observed ($Ho$), and the expected heterozygosity ($H_E$) for every locus were estimated. Deviations from Hardy-Weinberg equilibrium (HWE) were calculated for each locus for each sampling locality. Similar analyzes were done for COI, including pairwise $\Phi st$ index, haplotype (h) and nucleotide ($\pi$) diversity. In all cases, significance (*P*-values) were calculated using 10,000 non-parametric permutations. *P*-values were adjusted using Bonferroni correction for multiple comparisons (*Rice, 1989*).

Additionally, isolation by distance (IBD) was analyzed for COI and microsatellites, using the Mantel test from a matrix of genetic (*Fst*/ $\Phi st$; previously calculated) and geographic (km) distances among the sampled localities, using the Arlequin v3.5

software (*Excoffier & Lischer, 2010*) and following all steps included in its manual (http://cmpg.unibe.ch/software/arlequin35/man/Arlequin35.pdf). The geographical distances were obtained following the course of the rivers *via* the coordinates of each location using GoogleEarth Pro (Google Inc). The analyses were performed to evaluate if the spatial processes are driving population structure, or, in other words, if the IBD is the best explanation to the observed genetic structure patterns for the sedentary *P. axelrodi.*

To estimate the number of genetic clusters and to determinate the degree of admixture and gene flow among populations, we used the Bayesian clustering approach as implemented in the software STRUCTURE v2.3.3 (*Pritchard, Stephens & Donnelly, 2000*). We used the admixture and correlated frequencies priors and performed 10 replicates for $K = 1$ to 8 (testing each locality as a possible cluster). Assignment analyses were executed with 1 million steps Markov-Chain-Monte-Carlo (MCMC) with a burn-in of 10%. The convergence of the MCMC was inferred from the plot of $\alpha$ value of each independent run. The runs were analyzed in the program Structure Harvester v0.6.92 (*Earl & VonHoldt, 2011*), and independent replicates for each K were summarized in the program CLUMPP (*Jakobsson & Rosenberg, 2007*) and visualized using the software Distruct (*Rosenberg, 2003*). The most likely number of biological groups (K) was inferred using the method of *Evanno, Regnaut & Goudet (2005)*. However, since DeltaK (Evanno) leads to wrong inferences on hierarchical structure and downward-biased estimates of the true number of sub-populations, we used alternatively the Puechmaille method (*Puechmaille, 2016*) to infer the number of biological groups (K), using the web-based software StructureSelector (*Li & Liu, 2018*).

Patterns of genetic structure were further explored, using the diploid genotypes of 8 loci (16 variables) in the complete genotype matrix of the 163 individuals; through a discriminant analysis of principal components (DAPC) using R package Adegenet (*Jombart, 2008*), and following the instructions found in the manual developed by *Jombart & Collins (2015)*. DAPC construct linear combinations of the original variables (alleles) which have the largest between-group variance and the smallest within-group variance. First, the genotype matrix (STRUCTURE format) was converted to a *genind* object using the function "read.structure". Then, it was implemented the function "dapc", which first transforms the data using PCA, and then performs a Discriminant Analysis on the retained principal components.

According to the observed genetic structure patterns and biological groups formation through STRUCTURE and DAPC analyses, we defined biogeographical populations, and then, quantified the gene flow in terms of the effective number of migrants per generation (*Nm*) between theme, using the program Migrate v3.6.11 (*Beerli, 2006*), following the spirit of the methodology previously implemented by *Lagostina et al. (2018)*. For this purpose, we tested different migrations models between the paired predefined populations (1:full (xxxx); 2:unidirectional (x0xx); 3:unidirectional (xx0x); 4:panmictic (x)), and then, we selected the best one by comparing their Marginal Likelihood based on the Bezier approximation score, through the Bayes Factor method. For the Theta and M parameters calculations, we implemented uniforms priors. The search strategy used static heating scheme with four chains at different temperatures (1.0, 1.5, 3.0, 1000000.0) and one replicate of one long
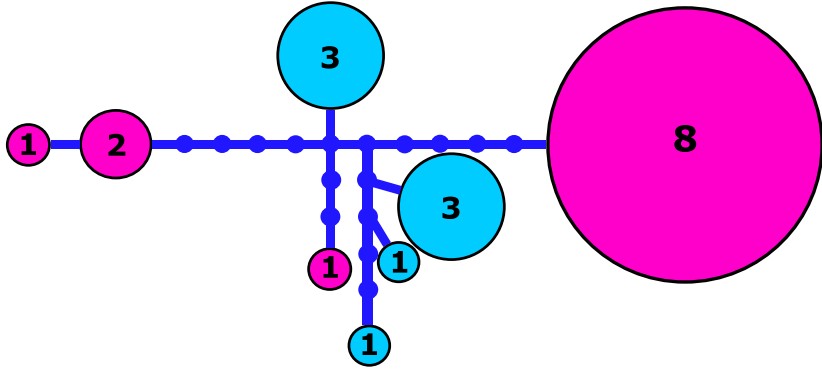

**Figure 2  Mitochondrial COI haplotype network.** Mitochondrial *Cytochrome oxidase I* (COI) haplotype network obtained from RAxML. The size circles reflect frequency of a particular haplotype in Orinoco and Negro basins. Inside every circle is shown the number of individuals sharing one haplotype. The dots indicate a number of mutational steps.

chain with 1 million generations, discarding a burn-in of 100,000, and sampling every 500 generations, as previously described in *Muniz et al. (2017)*. We diced to evaluate (1) the effective sample size (ESS >500), (2) autocorrelation values (<0.2), and, (3) the correct distribution of the probability density plots for estimates of the parameters; to verify the convergence of parameter estimates, as previously described in *Muniz et al. (2017)* and in *Lagostina et al. (2018)*. To convert the scaled parameter ''M'' to biological data in terms of number of effective migrants (Nm), we assumed a conservative generation time of 1 year (https://animaldiversity.org/accounts/Paracheirodon_axelrodi/); and a conservative substitution rate for fish microsatellites of $1.0 \times 10^{-5}$ mutations per locus per generation (reviewed in *Chistiakov, Hellemans & Volckaert (2006)*).

## RESULTS

### Phylogeographic patterns

We identified 2–4 haplotypes per locality, reaching 20 in total (see Table S1 and GenBank accessions MG384556–MG384575). The COI haplotype network did not show shared haplotypes between the Orinoco and Negro basins. However, no reciprocal monophyly was found between basins (Fig. 2; Fig. S1). In addition, for the 20 nuclear MYH6 gene sequences (see Table S1 and GenBank accessions MG384576–MG384595), they were inferred 40 alleles, and the resultant network showed 25 alleles shared among populations between basins (Fig. 3; Fig. S2), mainly for individuals from the Cucui and Santa Isabel with individuals from the Orinoco basin, breaking the reciprocal monophyly between basins (Fig. S2).

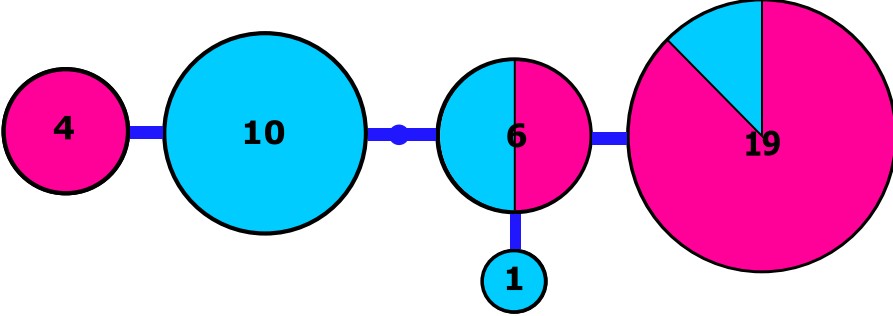

**Figure 3 Nuclear MYH6 gene allele network.** Nuclear myosin heavy polypeptide 6 cardiac muscle alpha (MYH6) gene alleles network obtained from RaxML. The size circles reflect frequency of a particular haplotype/allele in Orinoco and Negro basins. Inside every circle is shown the number of individuals sharing one haplotype. The dots indicate a number of mutational steps.

The analysis of phylogenetic reconstruction Relaxed Random Walk (Fig. 4; Fig. S3) estimated that *P. axelrodi* ancestral population was located in the Orinoco basin in the Late Pleistocene (*ca.* 0.255 Ma), between Inírida and Cucui, on the Vaupés Arch near the border of the Guiana Shield. The signal of both markers suggested that from there (approximately 0.115 Ma), that ancestral population of *P. axelrodi* dispersed faster along the Orinoco basin reaching all localities studied simultaneously. Later, around 0.019 Ma, it colonized the Negro River basin *via* Cucui, arriving in its middle basin through Santa Isabel in the Holocene, approximately between 0.003 Ma and 0.001 Ma. Finally, *P. axelrodi* reached the lower Negro River through Barcelos, the last location, to be colonized in the last 1,000 years.

## Gene flow and population genetic structure

FreeNA results showed that no null alleles were present in any of the loci analyzed. Genetic diversity values such as expected (*HE)* and observed Heterozygosity (*Ho)*, and number of alleles per population (n) were obtained from eight loci screened in all individual of *P. axelrodi* along with deviations from HW equilibrium. Microsatellite loci had between 2 and 22 alleles per locus. Heterozygosity values were very similar for Puerto Carreño and Puerto Gaitan while *HE* was the highest in Cucui and lowest in Barcelos (Table S2), after Bonferroni correction (*P-value* = 0.0007812) deviation from HWE was observed at the majority of loci in Orinoco basin populations except *(Pa7)*. Among the Negro basin populations, deviation from HWE was observed most loci except *(Pa7, Pa13, and Pa26)* (Table S2).

The $F_{ST}$ pairwise population differentiation analysis showed that within Orinoco basin there was no differentiation between Puerto Gaitán and Puerto Inírida (*F ST :* ∼0.010;

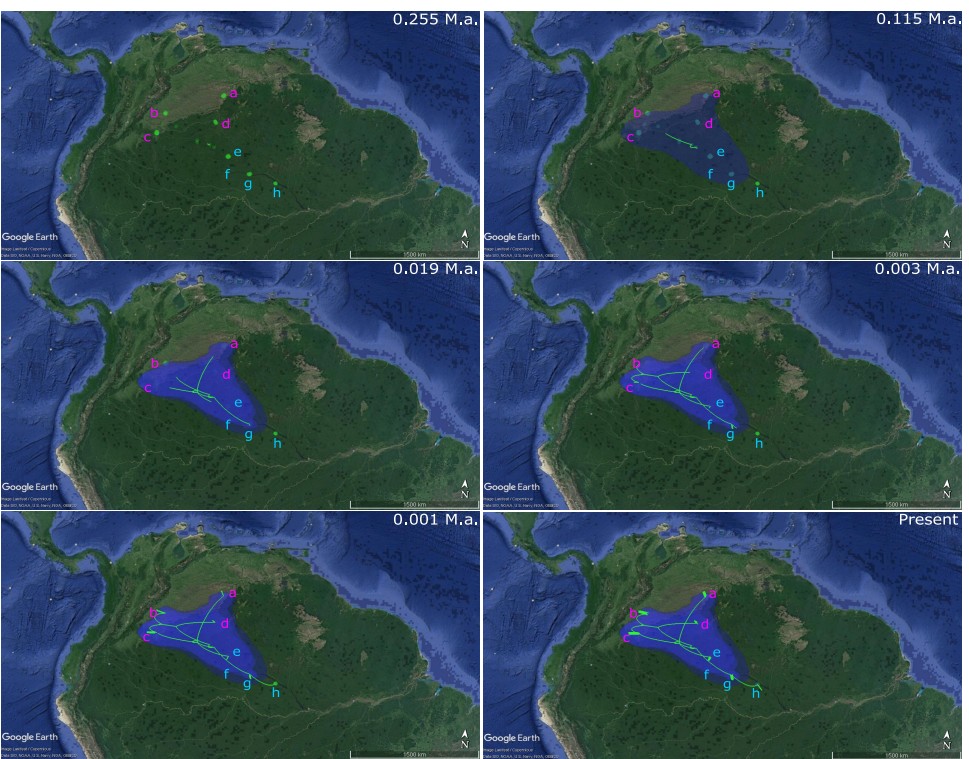

**Figure 4 Phylogeographical reconstruction analysis using non-concatenated dataset of the mitochondrial COI and the nuclear MYH6 gene, based on a relaxed random walk in continuous space and time method for *Paracheirodon axelrodi* from Orinoco and Negro basin, inferred in BEAST v. 1.8.2 and SPREAD3 and visualized in Google Maps.** Graph is showing the hypothetical spatial patterns and times of colonization for *P. axelrodi* in each basin, locating the ancestral population from all of them was originated. All the observed colonization times are median heights into 95% credibility intervals, which can be found in the supporting information (Fig. S1).

$P > 0.05$), showing panmixia between them. There was significant differentiation among the rest of populations with values ranging from ∼0.10 to 0.24; $P < 0.05$), showing structuring among them (Table S3). Additionally, inside Negro basin all populations were found to be differentiated ($F_{ST}$ values from ∼0.03 to 0.17; $P < 0.05$). According to the AMOVA, most of genetic variation in *P. axelrodi* was within populations (86.53%; $P > 0.05$). However, a small but significant portion (13.47%; $P < 0.0001$) was occurring among populations.

The pairwise $\Phi_{st}$ index for COI showed a high degree of significant structuring, except among some Orinoco populations ($\Phi_{st} = ∼0.05-0.115$; $P > 0.05$). For instance, in the Negro basin, each location was highly structured, showing an $\Phi_{st}$ index between ∼0.68 and 0.95 ($P < 0.05$). Similarly, pairwise differentiation values between basins, they ranged from 0.58 up to 0.98, indicating a highly significant structure ($P < 0.05$) (Table S4; Supplemental Information).

According to the Mantel Test, the correlation between pairwise $F_{st}$ and geographic distances for the eight *P. axelrodi* populations was moderate but statistically significant

($r = 0.591787$; $P = 0.008$). However, no robust evidence was found to infer Isolation by Distance, since only 35% of genetic differentiation observed among *P. axelrodi* populations can be explained by the geographic distance ($R^2 = 0.350211$).

The assignment test analysis conducted in STRUCTURE showed two biological clusters ($K = 2$) as the highest posterior probability of population structure (Mean Ln(K) = $-5063.080$), according to the Evanno method (Fig. 5; Fig. S4). The two populations units detected by STRUCTURE were: (1) Orinoco River populations (Guaviare, Puerto Carreño, Puerto Gaitan, and Puerto Inirida) + Cucui ($n = 103$); and (2) the remaining of the Amazon River populations (Barcelos, Santa Isabel and São Gabriel da Cachoeira) ($n = 60$) (Fig. 5). However, the Puechmaille method estimators suggested the existence of a genetic substructure, with $K = 6$ as the highest probability of biological clusters (Fig. S4). As an observable pattern from $K = 2$ to $K = 6$ (Fig. 5), the Rio Negro populations (except Cucui) are highly structured compared to those of the Orinoco. Cucui is shown as an independent population or more associated with the Orinoco River basin, but less associated with the Negro River basin. In all of them, Cucui and San Gabriel are observed as a transition zone between the Orinoco and Negro basins. The DAPC is concordant with the observed pattern $K = 2 - K = 6$ described above (Fig. 6). Both, STRUCTURE and DAPC showed Puerto Carreño (Orinoco basin) as the locality with the highest signals of co-ancestry or relatedness with the populations from the Negro River basin (except Cucui).

According to the genetic structure pattern observed, we tested in Migrate the gene flow among three biogeographical groups: Orinoco basin, Cucui and rest of Negro basin. As a result, we found that the unidirectional pattern was the best model explaining migration among these groups: Cucui toward Orinoco basin ($Nm = 4.8$ migrants per generation, Supplementary Material 1); Cucui toward rest of Negro basin ($Nm = 6.58$ migrants per generation, Supplementary Material 2) and rest of Negro basin toward Orinoco basin ($Nm = 4.5$ migrants per generation, Supplementary Material 3). Supplementary Material 1, 2, and 3 can be accessed through DRYAD, https://doi.org/doi:10.5061/dryad.866t1g1vv.

## DISCUSSION

### Genetic structure and phylogeographic patterns

Despite its sedentary condition and evidence of structuring between basins, cardinal tetra populations appear to have had a historical connection between basins. The nuclear alleles network and the genetic structure results, shown signals of this fact, and also, that Cucui (Negro basin) is more associated with the Orinoco basin, while Santa Isabel and Barcelos are shown as another independent biological group, being São Gabriel da Cachoeira the transition area between them. This observed pattern (Orinoco+Cucui / rest of the Negro basin) is probably explained by an ancient geological event such as the formation of waterfalls and rapids located in São Gabriel da Cachoeira. Our hypothesis suggests that the waterfalls may be the first split event dividing the Orinoco basin + Cucui populations from the rest of the Negro basin populations (São Gabriel de Cachoeira, Santa Isabel and Barcelos). It is known that the waterfalls system and rapids near São Gabriel da Cachoeira act as a physical barrier structuring the populations of small species between the upper

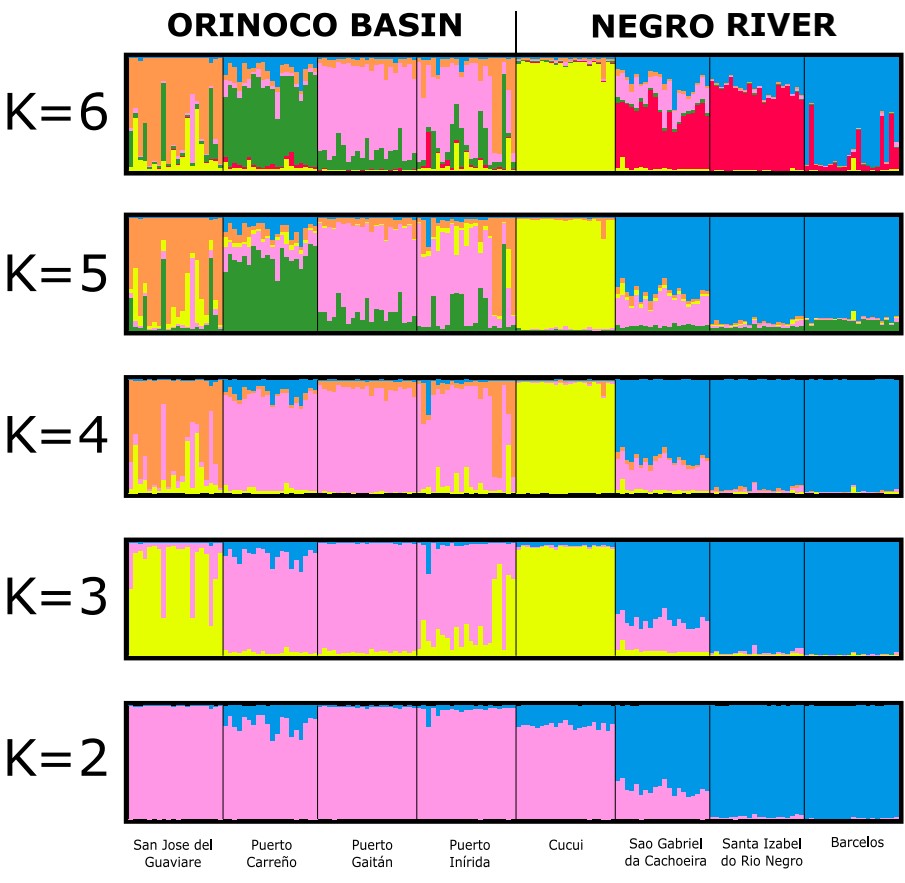

**Figure 5** **Bayesian assignment for each individual of *Paracheirodon axelrodi* inferred by the software Structure from $K = 2$ to $K = 6$, using eight microsatellite loci.** Vertical lines indicate each of the individuals and the colors represent the probability membership coefficient of that individual to each genetic cluster. GV, San José del Guaviare; CA, Puerto Carreño; PG, Puerto Gaitán; IN, Inírida; CUC, Cucuí; SGC, São Gabriel da Cachoeira; SI, Santa Isabel; BAR, Barcelos.

section and the remaining area of the Negro River (*Albert & Reis, 2011*; *Terencio, Schneider & Porto, 2012*).

However, when the results are explored hierarchically in STRUCTURE (K3–K6) and DAPC, it is found a substructure that shows Cucui isolated from the Orinoco, currently showing itself as an independent population from the Negro River and Orinoco River. This suggests that after the São Gabriel waterfalls formation, a second more recent split event occurred between the headwaters of the Negro River (where Cucui is located) and the Orinoco, until then connected probably through the Inírida River. This separation event was most likely influenced by the final Vaupes Arch rising. It is known that this rising happened from west to east, towards the Guiana Shield, and studies evidenced the influence of this process on fish fauna divergence between the resulting basins (*Mora et al., 2011*; *Caputo & Soares, 2016*). These split hypotheses modeling part of the structure of *P. axelrodi* between basins are also supported by the Mantel test, where the distance does not explain by itself the current patterns of genetic structuring.

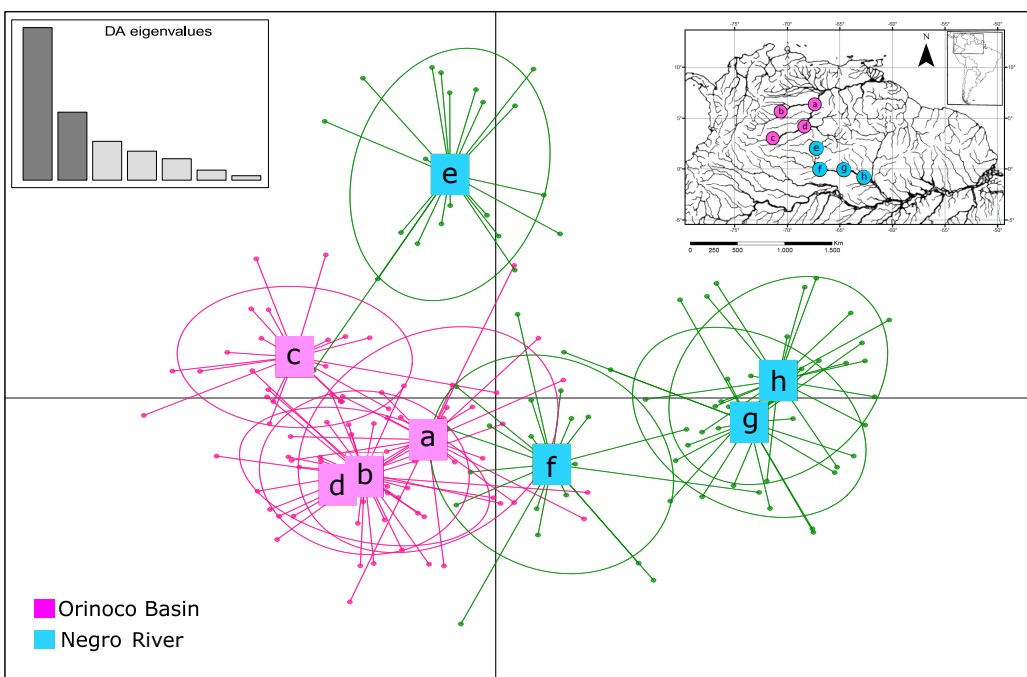

**Figure 6** **Patterns of genetic structure by discriminant analysis of principal components (DAPC)** *(Adegenet package in R)*, **using eight microsatellite loci for samples of** *Paracheirodon axelrodi* **from (A) Puerto Carreño; (B) Puerto Gaitán; (C) San José del Guaviare; (D) Inírida; (E) Cucuí; (F) São Gabriel da Cachoeira; (G) Santa Isabel; (H) Barcelos.**

Another of the remarkable genetic structure patterns was the found for Santa Isabel and Barcelos. STRUCTURE (K2- K6), DAPC and fixation indices, coincide in showing Santa Isabel and Barcelos as a genetically different stocks from São Gabriel da Cachoeira and Cucui. Since the Isolation by Distance hypothesis was rejected, the explanation for this pattern of genetic structure may be associated with the influence of the whitewater river in the section that comprises the distribution of these two populations or other of ecological order. This pattern of high genetic structure can also be observed in ornamental fish such as *Nannostomus eques* (*Terencio, Schneider & Porto, 2012*) and in *Carnegiella strigata* (*Schneider et al., 2012*). In these species, physicochemical differences in types of water produced by at least three whitewater tributaries such as: Demini, Padauari, and Branco Rivers and higher pH levels, sediment load, and turbidity, were showed as the more likely factors causing the observed structure in these populations. It is known that *P. axelrodi* is a blackwater fish (*Chao, 2001*; *Terencio, Schneider & Porto, 2012*), which allows it, as well as other native teleost to the Negro River, to have a special adaptation that allows them ionoregulatory specializations for exceptional tolerance of ion-poor, acidic waters of this blackwater river (*Gonzalez et al., 1998*; *Gonzalez & Wilson, 2001*; *Matsuo & Val, 2007*; *Wood et al., 2014*).

On the other hand, we observed that unlike the Negro River, there is not a very strong structure among the populations of the Orinoco basin. This can be explained by several characteristics of the area in which the Orinoco is distributed. For example, the

topography, being the Llanos region a flat area (savannahs (alluvial plain of the Orinoco River (0–50 m.a.s.l.)) (*Stallard, 1985*), which does not have relevant geological formations that may have caused a marked genetic vicariant or allopatric pattern of genetic structure in *P. axelrodi*. So, as there were no such strong barriers, this could have caused a more rapid, recent dispersal and establishment of populations, where there was not enough time to accumulate differences, as shown by Random Walk and pairwise $F_{ST}$ indices. However, according to STRUCTURE (K4–K6) and DAPC, we found genetic structure among Guaviare and the rest of the Orinoco populations. This and other structuring process ongoing for the Cardinal tetra, may be explained by three factors. First, since *P. axelrodi* is a blackwater fish (*Chao, 2001*; *Terencio, Schneider & Porto, 2012*), it is likely that Guaviare River, which is whitewater, can be a barrier preventing the current dispersion and connectivity of *P. axelrodi* with the rest of the Orinoco populations. Second, it is possible that the highest elevation of the Vaupes Arch in its western area (*Winemiller & Willis, 2011*; *Mora et al., 2011*), at the Andean Piedmont, where the Guaviare population is located (>200 m.a.s.l.), may be causing a gradual isolation of such individuals from the rest of Orinoco basin (mean altitude = 98 m.a.s.l.). Third, the life history and the small body size of this species can help prevent dispersion, generating structure (*Godinho, Lamas & Godinho, 2010*). This is expected for sedentary fishes such as *P. axelrodi*, especially between populations separated by large distances (*Lovejoy & Araujo, 2000*; *Cooke, Chao & Beheregaray, 2009*).

## Connectivity and dispersion routes

It is clear that the Vaupés Arch fragmented the transcontinental Paleo Amazonas-Orinoco River about ~5–10 Ma (*Hoorn et al., 1995*), generating until today 61.2% of exclusive taxa to Amazon and 16.6% to Orinoco (*Winemiller & Willis, 2011*). However, 22.2% of the taxa are still shared between basins until the present, probably by the persistent connections between the Orinoco and Amazon basins *via* the Casiquiare Canal and other blackwater tributaries (*Winemiller et al., 2008*). Until today, it was unclear to which of these three groups *P. axelrodi* belonged. However, as expected, in this study it was demonstrated that Cardinal tetra is a species still shared between Orinoco and Negro basins, establishing current (or at least until a very recent past) gene flow in a biological scale of ~4 migrants per generation between them. These results suggest a connection *via* Cucui toward Inírida route (*e.g.*, across western Casiquiare canal tributaries), or through a Cucui independent route, *via* rest of the Negro basin toward Orinoco (*e.g.*, across eastern Casiquiare canal tributaries, as suggested by the marked co-ancestry between Puerto Carreño and rest of the Negro basin). Patterns of connection between the Orinoco and Negro basins, has also been suggested previously in Cardinal tetra by *Cooke, Chao & Beheregaray (2009)*. Likewise, other previous genetic studies demonstrated that non-migratory fishes such as the cychlids *Cichla temensis*, *C. monoculus*, and *C. orinocensis* (*Willis et al., 2010*) or the black arowana *Osteoglossum ferreirai* (*Olivares et al., 2013*), experienced some degree of connectivity and gene flow between both basins, using the Casiquiare as their connection route. Contrary, aquatic species from white water environments such as the riverine turtle *Podocnemis unifilis* (*Escalona et al., 2009*) or the fish *Piaractus brachypomus* (*Escobar et al., 2015*), are

unable to currently establish connection/gene flow between the Orinoco and Amazon. It suggests that since these alternative connections are between blackwater drainages, dispersal between the Orinoco and Negro basins could be facilitated for small species that are adapted to blackwater environments (*Winemiller & Willis, 2011*), being *Paracheirodon axelrodi* one of such species.

There are currently at least five alternative connections among the tributaries of the upper Orinoco and Negro to the west of the Casiquiare canal (*Dungel, 2009*; *Winemiller & Willis, 2011*). One such connection is *via* the isthmus of Pimichin. The upper reaches of the Temi branch of the Atabapo River (Orinoco drainage) are connected with a 16 km marshy and swampy plain with the Pimichin creek, an affluent of the upper Guainia (Negro drainage) River. Another potential connection is *via* a Llanos floodplain, less than 10 km wide, that connects the left-bank affluents of the upper Guainia River and the Guacamayo creek that drain into the Inírida River (Orinoco basin). Yet another alternate connection is *via* the right bank affluent of the Conochirite River and tributaries of the Atacavi and Temi Rivers (Orinoco basin) separated by less than 10 km of flat, and at least seasonally marshy terrain. The Conochirite River is a left bank affluent of the Guainia River, and during the high-water season is also connected to the upper Casiquiare Canal. A fourth connection is between the floodplains of the Baria River (affluent of the Pasimony River, Casiquiare canal drainage) and the Maturaca River (affluent of the Cauaburi River, Negro River drainage). The connection *via* the Pimichin isthmus was already mentioned by Alexander von Humboldt and confirmed by *Rice (1921)*. Based on the examination of topographic maps, *Winemiller & Willis (2011)* suggest that this connection may have existed until very recently, or perhaps is current but ephemeral. The Conochirite River connection was also used by Humboldt instead of the lower Casiquiare Canal to arrive at the Guainia River near Meme da Maroa. The last potential connection is much further to the west across a less than 10 km flat floodplain between the upper Içana River, a tributary of the Negro River, and the Papunáua River, a tributary of the Inírida River.

Although the mitochondrial DNA did not show shared haplotypes between basins, this result is not conclusive to determine the absence of gene flow among Orinoco and Negro basins populations. This could mean that females are philopatric, a behavioral observation of males leaving the natal troop before sexual maturity and females staying with the troop for life (*Caparroz, Miyaki & Baker, 2009*). So, when a species has sex-biased dispersal, the simultaneous use of markers with different modes of inheritance (*i.e.,* uni- and biparental) can reveal the contrasting patterns of spatial distribution of the species' genetic variation. As a result, stronger population structure can be detected with the mtDNA than with nuclear DNA (*Caparroz, Miyaki & Baker, 2009*), such as the case of *P. axelrodi*.

However, there are other potential reasons for this result. Firstly, there were technical limitations in COI amplification for all 163 individuals in the study (uneven sampling (7–15 samples per locality)). In addition, localities such as São Gabriel da Cachoeira (with strong signals of connection with Orinoco (according microsatellites)) were not sampled for the COI marker due to low DNA quality, as explained above. Additionally, microsatellites are codominant markers that more robustly examine the demographic behavior of both genders in the population and not only the behavior of females (mtDNA) (*Freeland, 2005*).

## Historical biogeography and diversification patterns

In this study, according to mitochondrial and nuclear diversity indices, we found higher values in Orinoco River populations such as Puerto Inírida, Guaviare and Puerto Carreño. This may suggest that the ancestral origin of *P. axelrodi* is in the Orinoco basin, as reported by *Cooke, Chao & Beheregaray (2009)*, suggesting that Negro River headwater populations had the highest gene diversity values compared to the middle Negro River populations.

Random Walk analysis confirmed that ancient populations belong to the Orinoco basin, on the Western Guiana Shield, near Inírida. From here, it rapidly colonized the rest of Orinoco and almost simultaneously the upper Negro River (*via* Cucui), followed by the median and lower part of this river. These results are in accordance with the chronological Negro River formation proposed by *Latrubesse & Franzinelli (2005)*. These authors suggested that Negro River headwaters belong to an ancient formation while middle and lower Negro River floodplains during the final of Quaternary, belongs to a more recent formation. This observed headwater toward lower Negro colonization pattern, was also confirmed for *P. axelrodi* by *Cooke, Chao & Beheregaray (2009)*. These authors showed that populations in the headwater and upper sections of Negro River represent ancient lineages, while downstream populations represent more recent lineages.

Although the ancestral population of *P. axelrodi* was originated around 0.255 Ma, the colonization of the middle and lower Negro basin occurred only about <3,000 years ago. According to *Ruokolainen et al. (2018)*, the middle and lower sections of the Negro basin were not totally blackwaters until about 1,000 years ago, because of the direct influence of the whitewater Japurá River, that until then, drained into the Negro River, probably causing extensive changes in the physical-chemical characteristics of the water downstream middle Negro River, preventing the colonization of *P. axelrodi*, a restricted blackwater fish.

The colonization of *P. axelrodi* along the blackwater systems between the Orinoco and Negro basin, occurred rapidly and very recently, from the Late Pleistocene until the Holocene (~255,000–1,000 years ago; see Fig. S3), having its origin at the Western Guiana Shield. According to *Albert, Val & Hoorn (2018)*, during this period, it was evidenced a generalized pattern of river capture for the coastal-draining basins of the Western Guiana Shield. It was observed in the many signature elbows, U-turns, rapids and waterfalls of the uppermost Negro and Branco basin headwaters (*e.g.*, Casiquiare, Siapa, Mucajaí, Uraricoera and Taku-tu). These river captures resulted in geodispersal of coastal-basin headwaters to the Amazon (and probably to Orinoco), thereby enriching the Amazonian biota, and vicariances isolating sister species on either side of the Western Guiana Shield-Amazon divide (reviewed in *Albert, Val & Hoorn (2018)*). For instances, it is known that until about 1 million years ago, the upper Branco River drained into the Essequibo River (*Vale Schaefer Jr, 1997*), and only in the mid-late Pleistocene, about 0.25−0.1 Ma (*Cremon et al., 2016*), it was captured by the Negro River to become part of the Amazon basin.

## CONCLUSIONS

This study presents the first comprehensive analyses of population structure, connectivity, and phylogeographic patterns for cardinal tetra (*Paracheirodon axelrodi*) from the Orinoco

and Amazon basins using coalescent- and frequency-based analyses of mitochondrial and nuclear DNA markers. Our results suggest that *P. axelrodi* is a young species that originated at Western Guiana Shield drainages (near Inírida) at late Pleistocene (~0.255 Ma), with a very rapid and recent colonization of both basin from 0.115 Ma until about 1,000 years ago. Currently, the species is distributed between basins in a genetic structuring pattern of three geographical populations, even in the face of a restricted gene flow among them: (1) Orinoco (Guaviare, Puerto Gaitán, Puerto Inírida and Puerto Carreño), (2) Cucui (headwater Negro basin) and (3) rest of the Negro basin (São Gabriel da Cachoeira, Santa Isabel and Barcelos). Vaupés Arch is not a barrier to gene flow between basins for *P. axelrodi*. Instead, permanent connection between Orinoco and Negro basins, as the Casiquiare Canal or alternative black-water corridors drainages are allowing connection between both basins (currently or at least until a very recent past). This study provides key information to scientists, managers, and governmental agencies regarding the management of this fish as an important resource for ornamental industries in northern South American countries.

## ACKNOWLEDGEMENTS

The authors thanks Valéria Nogueira Machado, Mario Nunes and Sandra Hernández for their support in field and laboratory work.

### Funding

This study was supported by the Departamento Administrativo de Ciencia, Tecnología e Innovación (COLCIENCIAS) in a cooperation with the Conselho Nacional de Desenvolvimento Científico e Tecnologico (CNPq) of Brazil (code: 490682/2010-3); Laboratorio de Evolução e Genética Animal (LEGAL) from Universidade Federal da Amazonas UFAM and, by a Colombia Biodiversa grant from Fundación Alejandro Ángel Escobar and by the Proyecto semilla from Faculty of Science from the Universidad de los Andes of Colombia. The funders had no role in study design, data collection and analysis, decision to publish, or preparation of the manuscript.

### Grant Disclosures

The following grant information was disclosed by the authors:
The Departamento Administrativo de Ciencia, Tecnología e Innovación (COLCIENCIAS).
the Conselho Nacional de Desenvolvimento Científico e Tecnologico (CNPq) of Brazil (code: 490682/2010-3).
Laboratorio de Evolução e Genética Animal (LEGAL) from Universidade Federal da Amazonas UFAM.
A Colombia Biodiversa grant from Fundación Alejandro Ángel Escobar.
The Proyecto semilla from Faculty of Science from the Universidad de los Andes of Colombia.

## Competing Interests

The authors declare there are no competing interests.

## Author Contributions

- Diana Sanchez-Bernal conceived and designed the experiments, performed the experiments, analyzed the data, prepared figures and/or tables, authored or reviewed drafts of the article, and approved the final draft.
- José Gregorio Martinez conceived and designed the experiments, performed the experiments, analyzed the data, prepared figures and/or tables, authored or reviewed drafts of the article, and approved the final draft.
- Izeni Pires Farias conceived and designed the experiments, authored or reviewed drafts of the article, and approved the final draft.
- Tomas Hrbek conceived and designed the experiments, analyzed the data, authored or reviewed drafts of the article, and approved the final draft.
- Susana Caballero conceived and designed the experiments, authored or reviewed drafts of the article, and approved the final draft.

## Animal Ethics

The following information was supplied relating to ethical approvals (i.e., approving body and any reference numbers):

CICUAL Universidad de los Andes reviewed and approved protocols used. See attached letter.

## Field Study Permissions

The following information was supplied relating to field study approvals (i.e., approving body and any reference numbers):

Universidad de los Andes-Permiso Marco de Investigación Científica. In Brazil, permission to collect tissue samples was granted by SISBIO/IBAMA (11325-1/662933 - 1/10/2007), and in Colombia by ANLA (to Universidad de los Andes).

## Data Availability

The DNA sequences are available at Genbank: (COI) MG384556–MG384575; (MYH6) MG384576–MG384595.

The microsatellite genotypes used in this study and the supplementary material are available at Dryad: Sanchez-Bernal, Diana et al. (2023), Phylogeography and population genetic structure of the cardinal tetra (Paracheirodon axelrodi) in the Orinoco basin and Negro River (Amazon basin): evaluating connectivity and historical patterns of diversification, Dryad, Dataset, https://doi.org/10.5061/dryad.866t1g1vv.

## Supplemental Information

Supplemental information for this article can be found online at http://dx.doi.org/10.7717/peerj.15117#supplemental-information.

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
