# Peer review of "Phylogeography and population genetic structure of the cardinal tetra (Paracheirodon axelrodi) in the Orinoco basin and Negro River (Amazon basin): evaluating connectivity and historical patterns of diversification"

_PeerJ, doi:10.7717/peerj.15117_

## Round 0.1 · original submission · Minor Revisions

I received three thoughtful reviews of your paper. The general consensus is that only minor changes are needed; however, please pay close attention to the suggested revisions and comments of reviewer #2. I look forward to your reading your revised manuscript.

Reviewer 1 ·

Basic reporting

The language is clear and unambiguous throughout the manuscript. The article provides a strong introduction and background to demonstrate how the work fits into the broader field. Relevant prior literature is appropriately referenced. The structure of the article is appropriate. Figures are relevant to the content of the article, of sufficient resolution, and appropriately described and labeled. All appropriate raw data have been made available in accordance with our Data Sharing policy.

Experimental design

All methods are standard for the field. The research topic is similar to other papers published in the journal. The research question is well defined, relevant and meaningful. The investigation was and conforms to high technical and ethical standards. The methods are all described with sufficient information to be reproducible by another investigator.

Validity of the findings

The results are all reasonably interpreted within light of the methods, and the discussion is consistent with the results.

Additional comments

NA

Reviewer 2 ·

Basic reporting

Text mostly clear and unambiguous. Few grammar and structure problems that can be easily addressed by proof reading.

Literature references can be improved, mostly in the methods, introduction and discussion regarding ecological and biogeographical inferences.

Figures can be improved, especially the haplotype networks. I think the authors could also provide an additional phylogeny, resulting from their RAxML analysis. Tables can be added to complement the materials.

Results can be edited for clarity and cohesiveness.

Experimental design

Methods lack proper details.

Materials presented in a confusing way. Can be edited for clarity.

Validity of the findings

Findings seem to be par with results, and can be improved by improving some of the methods used. Comments below.

Additional comments

## Title: I would suggest choosing either Negro or Amazon, since it can be confusing for a reader that is not familiar with these basins

## Abstract
l23-24: The way this sentence is presented, it can be interpreted that there are assemblages of fishes shared between the Orinoco and Amazon that are not found in the Neotropics. Edit for clarity

l31: Please standardize the nomenclature of the genes: COI // MYH6 or CoI // Myh6 or coI // myh6. There are also inconsistencies throughout the text. Change pb to bp.

## Introduction

l72-75: Provide references for water chemistry properties and river origins. I suggest Sioli, 1984 and Crampton, 2011

l101-112: This is a good literature review of molecular studies of P. axelrodi, however it is disconnected from the previous sessions and the concluding paragraph below. I would suggest edits to highlight the knowledge gap that exists and is going to be addressed in this paper.

Materials and Methods:
l124: "DNA muscle tissue samples" makes no sense. Edit for clarity

L126: Is samples here referring to the fish, to the muscle tissue or the DNA datasets? Edit for clarity.

l123-135: The reader would benefit from a table that shows vouchers used, which of them are represented in each dataset and if there is any overlap.

l199: Change pb to bp.

l200: Where the sequences aligned manually? Or was an alignment algorithm used?

l201: How were haplotypes determined and how was the network inferred?

l207: As a GTR+GAMMA evolution model usually is fit for complex datasets, it can also over parameterize analyses. I do not believe choosing this model because the manual of a program suggest is a good approach. Evolution models are easily inferred by programs such as JModelTest, and are computationally fast, especially for datasets that are not massive. I recommend the authors to perform a proper inference of evolution model for their datasets. Also, were these analyses performed on a concatenated dataset or for individual genes?

l223: For the tree prior, was the tree rooted? And where?

l249: Mantel tests are tools that investigate correlations between two distance matrices, and can be used to infer isolation by distance. Please edit this session providing the rationale of using a mantel test, how it was performed and the steps.

l255: What is the rationale of k=1 to 6?

l261: While the Evanno method for deltaK inference is a thorough approach used in population genetics, it has its caveats. I would suggest the authors investigate further its applicability, especially referring to Puechmaille, 2016; Janes et al. 2017 and Cunningham et al. 2020. I think this is a necessary step once the optimal K recovered is 2, and k=2 can be related to algorithm biases.

l264: Please provide details on the steps taken in the DAPC analyses.

## Results

l288: How have the haplotypes been inferred and how was the network generated?

l289: How was reciprocal monophyly deliniated here? Lineages have to be properly defined for this inference to be made. Please provide more details.

l290-291: I would suggest providing a figure and a table with more details for the haplotype network. The numbers presented do not show the claims made by the authors in text.

l292-300: Since all these ages are based on an a priori evolutionary rate prior, I am curious if both markers would present similar results if analysed individually. I would also suggest the authors investigate paleogeographic patterns of connections of these basins for the recovered dates.

l314: Fst = 0.010 is not the same as Fst = 0.00, which would mean no differentiation. Please refine text to accurately reflect the results obtained. Are these values for microsat only?

l315-318: What test has been used to access significancy of the differentiation here? I would suggest straightforward tests like G-statistic permutations and bootstrapping permutation over loci. Why did the authors consider the Negro populations to be all different with an Fst of 0.03 but not the Orinoco ones with 0.01?

l331-333: Please refer to my previous comment regarding the use of the Evanno method for deltaK determination and the possible caveats of this approach.

## Discussion

l362-380: Please provide geological dates for the hypothesized paleogeographic events that could explain the patterns observed.

l370: If the results suggest K=2 (see my comment above about this inference) as the best K for the data, further exploration of the K3-K6 is rather redundant and somewhat circular. By forcing extra partitioning of the dataset, it is expected that independent groups would appear. Do they mean anything, if your tests show that the extra partitioning is not the best explanation for your dataset? The authors should rely on their results, or explain why this extra (and in theory not optimal) partitioning is relevant.

l384: Mantel is not a hypothesis, it is a test. Please edit for clarity.

l386: Any literature on this possible ecological isolation for P. axelrodi? It would benefit the manuscript if the authors could include references on the biology of this fish that could be linked to ecological isolation.

l395: Provide references on the geological and topographical information provided.

l402: This information about the ecology of P. axelrodi could be presented earlier to back previous claims about ecological isolation.

l460: This paragraph contradicts many claims made by the authors beforehand.

l467-469: Provide references

Reviewer 3 ·

Basic reporting

No comment.

Experimental design

No comment

Validity of the findings

No comment

Additional comments

Dear authors,

The review of the manuscript, with comments and suggestions, is in the pdf file.
This is a good contribution for Neotropical freshwater fishes.

Annotated reviews are not available for download in order to protect the identity of reviewers who chose to remain anonymous.

---

## Round 0.2 · accepted · Accept

Thank you for your thorough attention to the reviewers' comments and suggestions. Your paper is much improved because of your efforts.

---

## Author Rebuttal · Round 0.2

Bogotá, January 13, 2023

Dr. Suzanne Prange
Academic Editor
Dear Dr. Suzanne Prange,

We appreciate the constructive comments and suggestions provided by the editor and reviewers, which have led to improve our paper. To address their concerns, we have edited the manuscript following their recommendations.

We hope that the manuscript is now suitable for publication in *PeerJ.*

Yours sincerely,

Susana Caballero
Author for correspondence
On behalf of all authors

**Note:** To facilitate the evaluation of all tracked changes by the reviewers, we included in the rebuttal letter the new corrected paragraph, indicating the line number and highlighting the change in green colour.

# Responses for the Reviewer # 2

**Reviewer #2**:

-##Text mostly clear and unambiguous. Few grammar and structure problems that can be easily addressed by proof reading.

Done. We carry out a new proof reading.

-##Literature references can be improved, mostly in the methods, introduction and discussion regarding ecological and biogeographical inferences.

Done.

-##Figures can be improved, especially the haplotype networks. I think the authors could also provide an additional phylogeny, resulting from their RAxML analysis. Tables can be added to complement the materials.

Done. The haplotype networks (Figure 2 and 3) were reprinted in shapes and colors, and improved in their visual quality. Instead of JPG, we export it in high definition as a vector graphic (PDF).

Likewise, we generated the figures of the phylogenetic trees derived from the RAxML analyses (Figure S2 and S3), both for the COI haplotypes and for the MYH6 alleles. Additionally, we created Table S1, which reflects every detail associated with codes, names in the collection, depositories, haplotypes, geographic coordinates, etc., of each sample used.

-##Results can be edited for clarity and cohesiveness.

Done.

-##Methods lack proper details. Materials presented in a confusing way. Can be edited for clarity.

Done.

## Title: I would suggest choosing either Negro or Amazon, since it can be confusing for a reader that is not familiar with these basins.

We greatly appreciate the reviewer's suggestion, and he is right to consider that there may be confusion for those readers who do not know the Amazon. However, we also believe that it is our duty to educate and teach through the science we publish. Readers should learn that the Negro River is a tributary of the Amazon Basin. Thus, science will fulfill its duty to teach so that the reader is not confused in the future with similar readings in the area.

**Title now:**
"Phylogeography and population genetic structure of the cardinal tetra (*Paracheirodon axelrodi*) in the Orinoco basin and Negro river (Amazon basin): evaluating connectivity and historical patterns of diversification"

## Abstract

l23-24: The way this sentence is presented, it can be interpreted that there are assemblages of fishes shared between the Orinoco and Amazon that are not found in the Neotropics. Edit for clarity.

The reviewer is right.

**Now (Line 23-24)**:
"The Neotropics contain one of the most diverse assemblages of freshwater fishes worldwide. Part of this diversity is shared between the Orinoco and Amazon basins".

l31: Please standardize the nomenclature of the genes: COI // MYH6 or CoI // Myh6 or coI // myh6. There are also inconsistencies throughout the text. Change pb to bp.

Done. The nomenclature of the genes was standardized: COI // MYH6

## Introduction

l72-75: Provide references for water chemistry properties and river origins. I suggest Sioli, 1984 and Crampton, 2011

Done. We included Sioli (1984) and Crampton (2011).

l101-112: This is a good literature review of molecular studies of P. axelrodi, however it is disconnected from the previous sessions and the concluding paragraph below. I would suggest edits to highlight the knowledge gap that exists and is going to be addressed in this paper.

The reviewer is right. The changes are highlighted in green, as follows:

**Now (Line 99-121)**:
"Despite its extensive geographic distribution spanning two major South American river basins, and extensive commercial exploitation, very little is known about its population dynamics, population structuring or evolutionary history of this species using molecular markers.
        The first molecular study reported for *P. axelrodi* was that of Harris & Petry (2001). The authors analyzed the mitochondrial control region for specimens from two geographically proximate localities from the vicinity of Barcelos (Brazil) and specimens of unknown geographic origin and found 3.9% and 4.3% sequence divergence between these specimens. In a much more comprehensive study focusing on the Negro River, Cooke & Beheregaray (2007) found extremely high levels of genetic diversity in the S72 intron in specimens sampled from the tributaries of the Negro River. Subsequently, Cooke et al. (2009) evaluated population structure, history of colonization, and genealogical relationships, concluding that headwater populations showed higher genetic diversity than downstream populations. Bittencourt et al. (2017) analyzed temporal genetic variation within a single population using five microsatellite loci and concluded that changes in allelic frequencies over time are associated with droughts instigated by El Niño, although these events do not significantly reduce genetic diversity. Despite these studies, none has so far revealed the evolutionary behavior of the Orinoco population and its relationship to the Amazon populations, combining information from mitochondrial and nuclear markers.
        In this study we provide the first phylogeographic hypothesis and analysis of population structure for *P. axelrodi* in its entire geographic distribution in the Orinoco and Amazon basins using different molecular markers, including mitochondrial (COI), nuclear sequences (MYH6), and eight microsatellite loci. Additionally, we test the hypothesis of basins connectivity and provide support regarding potential routes of gene flow and colonization between the two basins".

Materials and Methods:

l124: "DNA muscle tissue samples" makes no sense. Edit for clarity

Done. The first paragraph of the Materials and Methods section was restructured to give it more meaning and clarity.

**Now (Line 127-140)**:
"A total of 163 muscle tissue samples (one per individual) were collected along the Orinoco and Amazon basins on the distribution area for *P. axelrodi*. The samples were collected by ornamental fish fishermen from four localities on the Orinoco basin (Colombia) including San José del Guaviare (n=20), Puerto Carreño (n=20), Puerto Gaitan (n=21), and Puerto Inirida (n=21) (Fig. 1), and four

localities in Negro River in the Amazon basin (Brazil) including Cucui (n=21), Santa Isabel (n=20), Barcelos (n=20), and São Gabriel da Cachoeira (n=20) (Fig. 1). Samples from Sao Gabriel da Cachoeira (Negro river – Amazon basin) were obtained from the Laboratório de Genética Animal – LGA at Instituto Nacional de Pesquisas da Amazônia - INPA. However, these samples could not be used for mitochondrial and nuclear DNA analysis due to the low integrity of DNA, but they were suitable for microsatellite analysis. Samples from all other Amazon localities were also obtained from local artisanal fishermen working in these areas selling ornamental fish. Tissue samples and DNAs were deposited on the Laboratorio de Ecología Molecular de Vertebrados Acuáticos – LEMVA at Universidad de los Andes, Colombia. Details on sample information and depositaries can be consulted in the Table S1".

L126: Is samples here referring to the fish, to the muscle tissue or the DNA datasets? Edit for clarity.

It was modified and clarified. See the answer above (or line 126 in the paper), first paragraph of the "*Sample collection and DNA extraction*" section from the "**Materials and Methods"**.

l123-135: The reader would benefit from a table that shows vouchers used, which of them are represented in each dataset and if there is any overlap.

The reviewer was right. So, we create the Table S1. All the information details about the samples used in this study were included.

l199: Change pb to bp.

Done.

l200: Where the sequences aligned manually? Or was an alignment algorithm used?

It was used an alignment algorithm (ClustalW). We included this and other details, as follows.

**Now (line 207-210):**
"All sequences were edited manually and aligned using the software Geneious v4.7 (Kearse et al., 2012). After the alignment by the ClustalW algorithm (Thompson, Higgins & Gibson, 1994), unique COI haplotypes within each population was inferred using the online fasta sequence toolbox DNAcollapser from FaBox v1.61 (https://birc.au.dk/~palle/php/fabox/; Villesen, 2007)".

l201: How were haplotypes determined and how was the network inferred?

See the answer above and the **Line 208**:
"After the alignment by the ClustalW algorithm (Thompson, Higgins & Gibson, 1994), unique COI haplotypes within each population was inferred using the online fasta sequence toolbox DNAcollapser from FaBox v1.61 (https://birc.au.dk/~palle/php/fabox/; Villesen, 2007)" …

In addition, see the **Line 211-226** (highlighted in green a new clarification about Haploviewer, the software used for infer haplotype network):
…"Then, a haplotype network was constructed for each gene to determine ancestry relationships. In the case of the MYH6, sequences were first phased into individual alleles using the PHASE algorithm (Stephens et al., 2001) incorporated into DnaSP v5.1 software (Librado & Rozas, 2009). Then, we used a maximum likelihood approach in RAxML v8.2.X for phylogenetic analyses (Stamatakis, 2014) in CIPRES (Miller et al., 2010) because of its ability to efficient and fast maximum likelihood tree search algorithm that returns trees with good likelihood scores. We used a GTR+GAMMA model of sequence evolution, as recommended by the authors of the program in the version 8.2.X manual, for single full ML tree searches, and 1,000 replicates of RAxML's rapid bootstrap algorithm to account for uncertainty in the estimation of the topology (Stamatakis et al., 2008). A consensus tree was constructed from the bootstrap output file without ruling out any of the 1,000 replicates, for the allele network visualization into Haploviewer software (Salzburger et al., 2011), a haplotype viewer program specifically developed for the purpose of reconstructing haplotypes networks using traditional phylogenetic algorithms, neighbour-joining, maximum parsimony and **maximum likelihood** genealogies from closely related, and hence, highly similar haplotype sequence data".

l207: As a GTR+GAMMA evolution model usually is fit for complex datasets, it can also over parameterize analyses. I do not believe choosing this model because the manual of a program suggest is a good approach. Evolution models are easily inferred by programs such as JModelTest, and are computationally fast, especially for datasets that are not massive. I recommend the authors to perform a proper inference of evolution model for their datasets. Also, were these analyses performed on a concatenated dataset or for individual genes?

The reviewer is right. What was written in the paper may have been misinterpreted because mistakes in the redaction. The author of the manual does not recommend to use the evolutionary model GTR+GAMMA because it is good or bad. The manual and creator of the RAxML program only includes GTR+GAMMA as the unique evolutionary model, so its selection is not optional when using this program (see the screen shot of RAxML on CIPRES, bellow).
Both the manual (https://cme.h-its.org/exelixis/resource/download/NewManual.pdf) and the CIPRES platform, includes GTR+GAMMA as the exclusive evolutionary model to run RAxML. In fact, the author explains and justify why it is used in RAxML to infer maximum likelihood genealogies: "*The GTRCAT approximation is a computational work–around for the widely used General Time Reversible model of nucleotide substitution under the Gamma model of rate heterogeneity. CAT servers the analogous purpose, that is, to accommodate searches that incorporate rate heterogeneity" … "The main idea behind GTRCAT is to allow for integration of rate heterogeneity into phylogenetic analyses at a significantly lower computational cost (about 4 times faster) and memory consumption (4 times lower)*".

Advanced Parameters

Nucleic Acid Options

Choose model for bootstrapping phase ○[Not Mandatory] ⦿GTRCAT ○GTRGAMMA

Protein Analysis Options

Choose GAMMA or CAT model: + ○Protein GAMMA ⦿Protein CAT
Protein Substitution Matrix + DAYHOFF ⌄

Upload a Custom Protein Substitution Matrix ⌄
Use a Partition file that specifies AA Matrices ☐
Select the First Protein Substitution Matrix Called in Your Partition File ⌄
Select the Second Protein Substitution Matrix Called in Your Partition File ⌄
Select the Third Protein Substitution Matrix Called in Your Partition File ⌄
Select the Fourth Protein Substitution Matrix Called in Your Partition File ⌄
Select the Fifth Protein Substitution Matrix Called in Your Partition File ⌄
Use empirical frequencies? ⦿No ○Yes

However, for our purposes about haplotype network reconstruction, the impact of the evolutionary model chosen is minimal, since what was intended was to obtain an ML genealogy that would allow evaluating whether or not haplotypes were shared between the basins.

RAxML has already been used successfully to obtain haplotype genealogies that were used to construct haplotype networks in Haploviewer, in order to detect gene flow by haplotype sharing (Martínez et al 2022, Molecular Phylogenetic and Evolution, 10.1016/j.ympev.2022.107517).

Regarding the use of JmodelTest, it was already included in the paper from the first version submitted (**see line 236-238**), since contrary to RAxML, BEAST does allow the inclusion of several evolutionary models.

Regarding the question: Were these analyses performed on a concatenated dataset or for individual genes? Haplotypes networks were done for individual genes. The phylogeographic reconstruction was done using non-concatenated dataset, but combining the phylogenetic signal of both genes. It is explained in the **Line 211 and 227**.

l223: For the tree prior, was the tree rooted? And where?

No. It was not rooted.

l249: Mantel tests are tools that investigate correlations between two distance matrices, and can be used to infer isolation by distance. Please edit this session providing the rationale of using a mantel test, how it was performed and the steps.

The reviewer is right. We rewrite the whole paragraph, as follows.

**Now (Line 265-273)**:
"Additionally, isolation by distance (IBD) was analyzed for COI and microsatellites, using the Mantel test from a matrix of genetic ($Fst/\Phi st$; previously calculated) and geographic (km) distances among the sampled localities, using the Arlequin v3.5 software (Excoffier & Lischer, 2010) and following all steps included in its manual (http://cmpg.unibe.ch/software/arlequin35/man/Arlequin35.pdf). The geographical distances were obtained following the course of the rivers via the coordinates of each location using GoogleEarth Pro (Google Inc). The analyses were performed to evaluate if the spatial processes are driving population structure, or, in other words, if the IBD is the best explanation to the observed genetic structure patterns for the sedentary *P. axelrodi*".

l255: What is the rationale of k=1 to 6?

We actually did two runs in parallel. In the first one we test for K=1-8, to test each locality as a possible cluster. However, K=7-8 did not show changes with respect to the pattern observed in K=6. At the same time, we performed a second run only for K=1-6, based on preliminary analysis patterns observed by AMOVA and DAPC.

In the end, the authors decided to show K=2-6, as it was less redundant, more specific, and consistent with the other analyses, including Puechmaille estimators (no shown in the first version submitted, but included in this new version), which detected K=6 as the most probable number of clusters (see Figure S4).

However, we rewrite the paragraph to include the analysis from K=1 to K=8, and the Puechmaille analysis, as follows.

**Now (Line 276-288):**
"We used the admixture and correlated frequencies priors and performed 10 replicates for K = 1 to 8 (testing each locality as a possible cluster). Assignment analyses were executed with 1 million steps Markov-Chain-Monte-Carlo (MCMC) with a burn-in of 10%. The convergence of the MCMC was inferred from the plot of α value of each independent run. The runs were analyzed in the program Structure Harvester v0.6.92 (Earl & Vonholdt, 2011), and independent replicates for each K were summarized in the program CLUMPP (Jakobsson & Rosenberg, 2007) and visualized using the software Distruct (Rosenberg, 2004). The most likely number of biological groups (K) was inferred using the method of Evanno et al. (2005). However, since DeltaK (Evanno) leads to wrong inferences on hierarchical structure and downward-biased estimates of the true number of sub-populations, we used alternatively the Puechmaille method (Puechmaille, 2016) to infer the number of biological groups (K), using the web-based software StructureSelector (Li & Liu, 2018)."

l261: While the Evanno method for deltaK inference is a thorough approach used in population genetics, it has its caveats. I would suggest the authors investigate further its applicability, especially referring to Puechmaille, 2016; Janes et al. 2017 and Cunningham et al. 2020. I think this is a necessary step once the optimal K recovered is 2, and k=2 can be related to algorithm biases.

Done. We included the Puechmille estimators.

l264: Please provide details on the steps taken in the DAPC analyses.

Done. The original paragraph was modified, as follows.

**Now (Line 289-298)**:
"Patterns of genetic structure were further explored, using the diploid genotypes of 8 loci (16 variables) in the complete genotype matrix of the 163 individuals; through a discriminant analysis of

principal components (DAPC) using R- package Adegenet (Jombart, 2008), and following the instructions found in the manual developed by Jombart & Collins in 2015 (https://adegenet.r-forge.r-project.org/files/tutorial-dapc.pdf). DAPC construct linear combinations of the original variables (alleles) which have the largest between-group variance and the smallest within-group variance. First, the genotype matrix (STRUCTURE format) was converted to a *genind* object using the function "read.structure". Then, it was implemented the function "dapc", which first transforms the data using PCA, and then performs a Discriminant Analysis on the retained principal components".

## Results

l288: How have the haplotypes been inferred and how was the network generated?

It was explained above.

l289: How was reciprocal monophyly delineated here? Lineages have to be properly defined for this inference to be made. Please provide more details.

The reviewer is right in the interpretation of the paragraph. However, what is intended to say is that there was no reciprocal monophyly between the basins (not among localities), since the focus of our analysis is the connection between the two basins. Since each basin does not behave as a monophyletic group, it is stated that there is no reciprocal monophyly between them.
My concept of reciprocal monophyly here is not *sensu stricto* for predefined lineages of a pair of populations at some point after they diverge from an ancestral population (Neigel & Avise 1986, Avise & Ball Jr. 1990). Instead, my concept of reciprocal monophyly is based on the criteria for conservation units (ESUs), that have frequently been based on levels of reciprocal monophyly (Moritz 1994, De Queiroz 2007).

We modified the paragraph to improve the interpretation by readers, as follows.

**Now (Line 322-329):**
"We identified 2-4 haplotypes per locality, reaching 20 in total (see Table S1 and GenBank accessions MG384556 - MG384575). The COI haplotype network did not show shared haplotypes between the Orinoco and Negro basins. However, no reciprocal monophyly was found between basins (Fig. 2; Fig. S1). In addition, for the 20 nuclear MYH6 gene sequences (see Table S1 and GenBank accessions MG384576 – MG384595), they were inferred 40 alleles, and the resultant network showed 25 alleles shared among populations between basins (Fig. 3; Fig. S2), mainly for individuals from the Cucui and Santa Isabel with individuals from the Orinoco basin, breaking the reciprocal monophyly between basins (Fig. S2)".

l290-291: I would suggest providing a figure and a table with more details for the haplotype network. The numbers presented do not show the claims made by the authors in text.

Done. The Table S1 and the Figures S1 and S2 were created. See in supplementary information.

l292-300: Since all these ages are based on an a priori evolutionary rate prior, I am curious if both markers would present similar results if analysed individually. I would also suggest the authors investigate paleogeographic patterns of connections of these basins for the recovered dates.

The reviewer is correct in thinking that the mitochondrial and nuclear genes have differences in recorded ages, given their differential mutational rates. Studying the phylogeographic patterns of each gene separately was not the focus of our study, since in that case we would be studying the story of the individual gene and not of the species. According Szöllősi and Daubin (2012) "*each gene tree reflects a unique story, which is linked to species history, but often significantly differs from it*". However, the objective of our study was precisely to correct this bias associated with the story of a single gene, so we amplified one mitochondrial and one nuclear to obtain information from both genomes and tell the history of the species, using coalescent-based approaches in BEAST, which was designed to combine the signal from gene trees to build a tree that reflects the behavior of the species.

Regarding the paleogeographic information associated with the times obtained (0.25 Ma up to 1000 years), it is very scarce for this Orinoco/Amazon connection region, specially about upper Orinoco basin. Most of the paleogeographic information that can be found for these very recent ages was found in gray literature, but not always associated with the upper Orinoco (Inírida) and upper Río Negro (Cucui). However, we review all the literature on the matter, including the most recent geological and paleogeographic review on the matter, made in 2018 by the scientists who have most rigorously studied this region: James Albert and Carina Hoorn (The changing course of the Amazon River in the Neogene: center stage for Neotropical diversification). In addition, we reviewed all the papers associated to Orinoco-Amazon connection (genetic and/or geology), including the Casiquiare region. All the scientifically reliable references that help to explain our phylogeographic patterns were included in the paper [See both "**Connectivity and dispersion routes"** section (line 459) and the last paragraph of "**Historical biogeography and diversification patterns**" section (line 540-560) in the **Discussion**).

l314: Fst = 0.010 is not the same as Fst = 0.00, which would mean no differentiation. Please refine text to accurately reflect the results obtained. Are these values for microsat only?

As explained in Materials and Methods, we used the Arlequin software to compute all the metrics associated with genetic differentiation indices, both with microsatellites (*Fst*) and mtDNA (*Φst*). We implemented the Exact Tests of Population Differentiation, which, according to the authors (See de manual: http://cmpg.unibe.ch/software/arlequin35/man/Arlequin35.pdf) is a test of non-random distribution of diversity into population samples under the **hypothesis of panmixia**.

Therefore, we not only estimate the differentiation index, but also test its significance through a hypothesis test. Thus, the null hypothesis (H0: Fst=0) is the existence of panmixia, while the alternative hypothesis (H1: Fst>0) is the existence of genetic structure. In other words, we are testing if our Fst/*Φst* metrics are statistically greater than 0 or not. Thus, in any hypothesis test, P values (probability of being wrong when rejecting the null hypothesis) that are less than 5% allow us to reject H0, while values greater than 5% do not allow us to reject H0.
From this rationale, when the value of P<0.05, the H0 is rejected and the existence of a genetic structure (differentiation) is concluded, while if P>0.05, it is concluded that there is no genetic structure (no differentiation).

Although mathematically 0.01 is not the same as 0.00, from a statistical point of view, they can be, according to hypothesis tests, which are based on probability.
This hypothesis has been tested successfully in other important genetic studies (See Cote et al. 2012, Molecular Ecology; Population genetics of the American eel (*Anguilla rostrata*): FST = 0 and North Atlantic Oscillation effects on demographic fluctuations of a panmictic species).

l315-318: What test has been used to access significancy of the differentiation here? I would suggest straightforward tests like G-statistic permutations and bootstrapping permutation over loci.

Arlequin software calculates the significance of the covariance components and fixation indices associated with the different possible levels of genetic structure (within individuals, within populations, within groups of populations, among groups) using non-parametric permutation procedures (Excoffier et al. 1992).

For clarity, we include a more specific description of the procedure, as follows.

**Now (Line 257-264):**
Arlequin v3.5 (Excoffier & Lischer, 2010) was used to run an Analysis of molecular variance AMOVA and Pairwise *Fst* among localities. Additionally, the number of alleles per locus (NA), the observed (Ho), and the expected heterozygosity (HE) for every locus were estimated. Deviations from Hardy-Weinberg equilibrium (HWE) were calculated for each locus for each sampling locality. Similar analyzes were done for COI, including pairwise *Φst* index, haplotype (h) and nucleotide (π) diversity. In all cases, significance (P-values) were calculated using 10,000 non-parametric permutations. P-values were adjusted using Bonferroni correction for multiple comparisons (Rice, 1989).

Why did the authors consider the Negro populations to be all different with an Fst of 0.03 but not the Orinoco ones with 0.01?

The decision to define differentiation or not among populations is not based on an arbitrary author criterion or the simple Fst/$\Phi st$ metrics, but on a statistical test. As explained above, the differentiation test ($Fst$ or $\Phi st$) is accompanied by a hypothesis tests of panmixia, which is based on a statistical probability. Reject or not the null hypothesis of panmixia is based on the P-value.

l331-333: Please refer to my previous comment regarding the use of the Evanno method for deltaK determination and the possible caveats of this approach.

Done.

**Now (Line 283-288):**
"The most likely number of biological groups (K) was inferred using the method of Evanno et al. (2005). However, since DeltaK (Evanno) leads to wrong inferences on hierarchical structure and downward-biased estimates of the true number of sub-populations, we used alternatively the Puechmaille method (Puechmaille, 2016) to infer the number of biological groups (K), using the web-based software StructureSelector (Li & Liu, 2018)".

## Discussion

l362-380: Please provide geological dates for the hypothesized paleogeographic events that could explain the patterns observed.

Regarding the paleogeographic information (dates) associated with the times obtained (0.25 Ma up to 1000 years), it is very scarce for this Orinoco/Amazon connection region, specially about upper Orinoco basin. Most of the paleogeographic information that can be found for these very recent ages was found in gray literature, but not always associated with the upper Orinoco (Inírida) and upper Río Negro (Cucui). However, we review all the literature on the matter, including the most recent geological and paleogeographic review on the matter, made in 2018 by the scientists who have most rigorously studied this region: James Albert and Carina Hoorn (The changing course of the Amazon River in the Neogene: center stage for Neotropical diversification). In addition, we reviewed all the papers associated to Orinoco-Amazon connection (genetic and/or geology), including the Casiquiare region. All the scientifically reliable references that help to explain our phylogeographic patterns were included in the paper (See "**Connectivity and dispersion routes"** and "**Historical biogeography and diversification patterns**" sections in the **Discussion**; line 459 and 523, respectively).

l370: If the results suggest K=2 (see my comment above about this inference) as the best K for the data, further exploration of the K3-K6 is rather redundant and somewhat circular. By forcing extra partitioning of the dataset, it is expected that independent groups would appear. Do they mean anything, if your tests show that the extra partitioning is not the best explanation for your dataset? The authors should rely on their results, or explain why this extra (and in theory not optimal) partitioning is relevant.

It was explained above. See the justification in the question "l255: What is the rationale of k=1 to 6?"

l384: Mantel is not a hypothesis, it is a test. Please edit for clarity.

The reviewer is right. We edit it.

**Now (Line 417-420):**
"Since the Isolation by Distance hypothesis was rejected, the explanation for this pattern of genetic structure may be associated with the influence of the white-water river in the section that comprises the distribution of these two populations or other of ecological order".

l386: Any literature on this possible ecological isolation for P. axelrodi? It would benefit the manuscript if the authors could include references on the biology of this fish that could be linked to ecological isolation.

Done. We included at least four new references supporting the ecological isolation (highlighted in green).

**Now (line 423-436):**
"This pattern of high genetic structure can also be observed in ornamental fish such as *Nannostomus eques* (Terencio et al., 2012) and in *Carnegiella strigata* (Schneider et al., 2012). In these species, physicochemical differences in types of water produced by at least three whitewater tributaries such as: Demini, Padauari, and Branco Rivers and higher pH levels, sediment load, and turbidity, were showed as the more likely factors causing the observed structure in these populations. It is known that *P. axelrodi* is a blackwater fish (Chao et al., 2001; Terencio et al., 2012), which allows it, as well as other native teleost to the Negro River, have a special adaptation that allows them ionoregulatory specializations for exceptional tolerance of ion-poor, acidic waters of this blackwater river (Gonzalez et al., 1998; Gonzalez & Wilson, 2001; Matsuo & Val, 2007; Wood et al., 2014)."

l395: Provide references on the geological and topographical information provided.

Done. We included Stallard (1985) [River Chemistry, Geology, Geomorphology, and Soils in the Amazon and Orinoco Basins] to support the savannahs concept and plain area for the Orinoco basin.

**Now (Line 437-442):**
"On the other hand, we observed that unlike the Negro River, there is not a very strong structure among the populations of the Orinoco basin. This can be explained by several characteristics of the area in which the Orinoco is distributed. For example, the topography, being the Llanos region a plain area [savannahs (alluvial plain of the Orinoco River (0–50 m.a.s.l.)] (Stallard, 1985), which does not have relevant geological formations that may have caused a marked genetic vicariant or allopatric pattern of genetic structure in *P. axelrodi*".

l402: This information about the ecology of P. axelrodi could be presented earlier to back previous claims about ecological isolation.

Done.

**Now (Line 432-436):**
"It is known that *P. axelrodi* is a blackwater fish (Chao et al., 2001; Terencio et al., 2012), which allows it, as well as other native teleost to the Negro River, have a special adaptation that allows them ionoregulatory specializations for exceptional tolerance of ion-poor, acidic waters of this blackwater river (Gonzalez et al., 1998; Gonzalez & Wilson, 2001; Matsuo & Val, 2007; Wood et al., 2014)".

l460: This paragraph contradicts many claims made by the authors beforehand.

We believe that, on the contrary, in the paragraph, the authors are explaining why there is a pattern of "no sharing of mitochondrial haplotypes" but "sharing of nuclear alleles (MYH6 and SSRs)".
The philopatry is one of our putative explanation to that pattern. So, when a species with sex-biased dispersal, the simultaneous use of markers with different modes of inheritance (uni- and biparental, as our case) can reveal the contrasting patterns of spatial distribution of the species' genetic variation.

Then, stronger population structure can be detected with the mtDNA than with nuclear DNA. This is in accordance with behavioral observations of males of several animal species leaving the natal troop before sexual maturity and females staying with the troop for life".

It is a phenomenon that generate discordant patterns between the mitochondrial and the nuclear signals (Caparroz et al. 2009; https://doi.org/10.1525/auk.2009.07183), as we observed in *P. axelrodi*. To improve the clarity for the readers, we modified the paragraph, as follows.

**Now (Line 506-514):**
"Although the mitochondrial DNA did not show shared haplotypes between basins, this result is not conclusive to determine the absence of gene flow among Orinoco and Negro basins populations. This could mean that females are philopatric, a behavioral observation of males leaving the natal troop before sexual maturity and females staying with the troop for life (Caparroz, Miyaki & Baker, 2009). So, when a species with sex-biased dispersal, the simultaneous use of markers with different modes

of inheritance (i.e., uni- and biparental) can reveal the contrasting patterns of spatial distribution of the species' genetic variation. As a result, stronger population structure can be detected with the mtDNA than with nuclear DNA (Caparroz, Miyaki & Baker, 2009), such as the case of *P. axelrodi*."

l467-469: Provide references

Done. We included Freeland (2005).

**Now (line 519-521):**
"Additionally, microsatellites are codominant markers that more robustly examine the demographic behavior of both genders in the population and not only the behavior of females (mtDNA) (Freeland, 2005)".

# Responses for the Reviewer # 3

All concerns associated to the grammar and symbols were correctly addressed.

Reviewer#3: ## Check also Albert et al. (2020).
https://www.annualreviews.org/doi/abs/10.1146/annurev-ecolsys-011620-031032

Done. The reference was included and the paragraph was modified, as follows.

**Now (line 53-57):**
"The Neotropics contain one of the most diverse assemblages of freshwater fishes in the world (Reis, Kullander & Ferraris, 2003; Turner et al., 2004; Albert, Tagliacollo & Dagosta, 2020). Within the Neotropics, the Orinoco and Amazon basins are characterized by their high ichthyofauna diversity, exceeding 9,000 total species, of which ~ 5,160 are freshwater species (Reis et al., 2016; Albert, Tagliacollo & Dagosta, 2020)".

Reviewer#3: ## Update with recent publications for Reis et al. (2016).

See above.

Reviewer#3: ## What about the voucher specimens of all analyzed fish, are there in any scientific collection? It is very important have vouchers in any fish collection.

Done. The Table S1 was created, which reflects every detail associated with codes, names in the collection, depositories, haplotypes, geographic coordinates, etc., of each sample used in this study.

In addition, we included information about the scientific collection.

**Now (Line 138-139):**
"Tissue samples and DNAs were deposited on the Laboratorio de Ecología Molecular de Vertebrados Acuáticos – LEMVA at Universidad de los Andes, Colombia. Details on sample information and depositaries can be consulted in the Table S1".